# Biomechanical Analysis on Skilled Badminton Players during Take-Off Phase in Forehand Overhead Strokes: A Pilot Study

**DOI:** 10.3390/biology11101401

**Published:** 2022-09-26

**Authors:** Xinze Cui, Wing Kai Lam, Qiang Gao, Xin Wang, Tianyu Zhao

**Affiliations:** 1Department of Kinesiology, Shenyang Sport University, Shenyang 110115, China; 2Sports Information and External Affairs Centre, Hong Kong Sports Institute, Hong Kong 999077, China; 3Hangzhou Xingduo Youchen Sports Technology Co., Ltd., Hangzhou 310000, China; 4Key Laboratory of Structural Dynamics of Liaoning Province, College of Sciences, Northeastern University, Shenyang 110819, China

**Keywords:** inverse dynamical analysis, lower limbs, propulsion, movement speeds

## Abstract

**Simple Summary:**

Different movement speeds can contribute to different joint loading in sports. Joint contact force is the actual force acting on the articular surface, which could predict performance and injury, but is rarely reported for badminton overhead strokes. From the perspective of sports biomechanics, this paper analyzes the characteristics of kinematics and mechanics at each stage of the fast and moderately paced movements and studies the changes in the force of the lower extremity joints caused by the characteristics of the movements at different speeds so that athletes and enthusiasts can clarify the essentials of the movements and prevent sports injuries.

**Abstract:**

Different movement speeds can contribute to different joint loading in sports. Joint contact force is the actual force acting on the articular surface, which could predict performance and injury, but is rarely reported for badminton overhead strokes. Through an approach using musculoskeletal modelling, six male elite badminton players performed forehand overhead strokes at different movement speeds (fast (100%) vs. moderate (90%)). The synchronized kinematics and ground reaction force (GRF) data were measured using a motion capturing system and a force platform. All kinematics and GRF information were input into the AnyBody musculoskeletal modelling to determine the three-dimensional hip, knee and ankle contact forces. Paired *t*-tests were performed to assess the significant differences among the GRF, joint kinematics and contact force variables between the movement speed conditions. The results showed that when compared with the moderate movement condition, participants performing faster stroke movements induced larger first and second vertical peaks and larger first horizontal peak but lower second horizontal peak, and it also led to higher peak ankle lateral and distal contact forces, knee lateral and distal contact forces, and hip distal contact forces. Additionally, fast movements corresponded with distinct joint angles and velocities at the instant of initial contact, peak and take-off among the hip, knee and ankle joints compared with moderate movement speeds. The current results suggest that changes in joint kinematics and loading could contribute to changes in movement speeds. However, the relationship between lower limb joint kinematics and contact forces during overhead stroke is unclear and requires further investigation.

## 1. Introduction

Badminton is one of the most popular sports in the world, and there are over 200 million participants playing at recreational and elite levels [1]. To attain performance proficiency, the players require a high level of technical skills, equipment, tactics and rapid footwork, including lunging, braking, turning, jumping and landing during game plays [2,3]. Efficient footwork allows participants to reach a shuttlecock at the best position quickly and to maintain good body balance for upper limb racket performance [4,5,6,7,8].

Previous studies have investigated the upper limb mechanics in various badminton-specific tasks [9,10,11]. Barnamehei et al. [11] identified significant differences in upper limb muscular activation and movement coordination between elite and less skilled players during badminton forehand smashes. Zhang et al. [9] found that trunk rotation would contribute higher shuttlecock release speed in a badminton forehand smash. Taha et al. [10] developed a virtual reality training system using Kinect Technology and inertial measurement unit (IMU) for players to learn upper limb mechanics (wrist, elbow and shoulder joint) of the smashing movement and showed improvements in the athletic performances. Koike and Hashiguchi [12] found that an increased racket head speed was related to increased joint torques and shaft-restoring torque of the racket arm during a badminton smash. However, while these studies have predominantly focused on performance and racket mechanics, little attention has been paid to the lower limb mechanics, which are also related to performance and injury.

In badminton, the demanding footwork and movement (e.g., lunging) could induce excessive and repetitive loading [6,7], which is associated with increased risk of joint injuries, stress fractures and micro-damage of cartilage [8]. During the London Summer Olympic Games 2012, about 11% and 7% of the athletes were reported as being injured and experiencing discomfort/illness, respectively [13]. Most of the injuries occurred in the lower extremities, especially the knee joint [14,15,16,17,18]. Forehand overhead stroke is one of the most frequently executed movements in badminton plays and often causes anterior cruciate ligament (ACL) and ankle injuries [19,20]. Therefore, an analysis of lower limb loading is necessary to understand the underlying mechanism of injuries in badminton. To date, previous studies have predominantly investigated the lower limb mechanics in forward lunge landing [4,6,7,21,22,23], but an investigation of overhead movement tasks is lacking. The knee joint kinematic and valgus moment of badminton players during the landing from overhead stroke [24] and high clear stroke [25] has also been reported. However, these studies did not report on the joint contact forces during the landing or stepping (push-off) phases. Joint moment is the extrinsic force that could be counteracted by intrinsic forces such as muscle forces, and therefore, it is not necessarily equivalent to the actual mechanical burdens on the articular joint interfaces [26]. Joint contact force is the actual force applied on the articular surface that combines the net joint reaction forces and forces generated by the muscles crossing the joint [27]. While Kimura et al. focused only on the landing phase after an overhead stroke, the take-off biomechanics remain unclear. The joint contact forces and/or take-off biomechanics (stepping step) can allow for better prediction on performance and injuries in various movement tasks [22,28,29].

The development of computational simulation platforms in engineering [30,31,32,33,34] allows for the joint contact force to be determined using musculoskeletal human modelling [35]. The modelling has been used to calculate the muscle forces and contraction velocity as well as joint contact forces for volleyball strokes [31], soccer instep passing [32] and badminton lunges [22,36]. Furthermore, it is still questionable how joint contact forces would change in maximum and moderate movement speed conditions. Hence, the objective of this study was to examine lower limb joint contact forces in forehand overhead strokes in maximum and moderate movement speeds. The results of this study can play an important role in injury prevention and performance in badminton overhead strokes.

## 2. Materials and Methods

### 2.1. Participants

According to the sample size calculation by Gpower 3.1 software (Informer Technologies, Inc., Los Angeles, CA, USA), six right-handed skilled badminton players (mean age = 25.0 ± 0.9 years, body height = 176.2 ± 1.2 m, body mass = 66.0 ± 3.2 kg) were recruited for this study. They were competitive club-level badminton players with regular training of at least ten years (ranged from 10 to 13 years). The participants were all national first-level badminton players for China. All of them were right-handed players and had no any injuries on their lower limbs, trunk and upper limbs for the past six months. To prevent fatigue, they were instructed not to engage in strenuous exercise within 24 h before the test. This study was received and granted by the ethical review board of Shenyang Sport University. All participants signed an informed consent form before testing.

### 2.2. Experimental Procedure

Participants performed all forehand overhead strokes (maximum and moderate speeds) at a biomechanical laboratory that was equipped with a synchronized motion capturing system (Codamotion, Charnwood Dynamics Ltd., Rothley, UK, sampling at 200 Hz) and a force plate (AMTI, Watertown, MA, USA, sampling at 1000 Hz) for the collection of lower limb kinematic and kinetics data. Participants wore the same badminton shoes (Victor SH-P7800, Victor Rackets Corp, Taiwan, China) to avoid potential bias due to footwear worn [37,38,39]. There were 18 Coda active markers attached firmly to the following anatomical landmarks: left and right sides of the first and fifth metatarsophalangeal joint, heel, lateral malleolus, lateral aspect of the shank and thigh segments, lateral epicondyle of the knee, as well as anterior superior iliac spine (ASIS) and posterior superior iliac spine (PSIS).

The footwork and overhead stroke movements were demonstrated by the same badminton coach to all participants (Figure 1). Prior to data acquisition, in order to make sure that the participants were familiarized with the experimental process and the deviation in the ground feeling during the actual test, three familiarization trials were conducted for each participant to become familiar with the experimental process before the data acquisition. The participants were given time for warm-up and familiarization with the movement and speed requirement. Upon the auditory signal, the participants were asked to perform one step to the back from the starting position and to land on their right foot on the force plate, followed by an immediate vertical jump for an overhead stroke with a racket at their maximum (fast) and submaximal (moderate) speeds. The participants returned to the starting position immediately after landing [24]. A shuttlecock was suspended 2.8 m (hanging height depended on the player’s body height) above the ground to simulate the average hitting point of the stroke [40,41,42,43]. A successful trial was defined as the right foot (stepping step) on the middle of the force platform and natural movement for the forehand overhead strokes with a racket in their right hand at the maximum (fast) and submaximal (moderate) movement speeds. Fast movement was defined as the fastest speed to complete the stroke movement with their maximum effort. Moderate movement was defined as 90% of the fast condition. The average fast and moderate movement speeds were 3.51 m/s ± 0.5 m/s and 3.17 m/s ± 0.8 m/s, respectively. The order of movement speed conditions presented were counterbalanced across all participants. Each participant performed three successful trials for each of the two movement speeds, resulting in a total of six trials for each participant. Both placing the markers and sounding the auditory signals were carried out by the same experimenters.

Spline interpolation was performed for minor missing data points in the Coda Motion (Codamotion cx1, Charnwood Dynamics Ltd., Rothley, UK) analysis kits. Only the take-off/stepping step (right leg) was analyzed in this study. The stance phase was defined as the period from touchdown to take-off (Figure 1). The instants of touchdown and take-off were determined when the vertical GRF first exceeded 10 N and reduced to 10 N, respectively. Kinematic data were filtered through a low-pass Butterworth filter with a cut-off frequency of 10 Hz as it is the standard procedure applied in the AnyBody Modelling System (AMMR5.0, AnyBody Technology A/S, Aalborg, Denmark). All kinematic and kinetic data were input into the musculoskeletal modelling (AnyBody Technology A/S, Aalborg, Denmark) for further processing. We used the AnyBody built-in generic model with seven rigid-body segments; six degrees of freedom; and a total of 159 musculotendinous units, which were scaled to the mass and anthropometries of each participant [22]. Inverse kinematics was used to determine the time-series joint angles that aligned with the stepping step of the overhead stroke movements. Dynamic inconsistency found in the Residual Reduction Algorithm was reduced by small adjustments to the model mass properties. The hip and ankle joints were modelled as spherical joints, while the knee joint was modelled as a hinged joint to predict the lower-limb joint contact forces. A positive value denoted dorsiflexion (knee/hip extension), pronation (valgus) and internal rotation for the respective joints.

The analyzed variables were joint angles and angular velocities during the take-off step (at initial contact, peak value and take-off); GRF; and peak sagittal, coronal and transverse hip, knee and ankle contact forces. These variables were selected as they are linked to badminton injuries and performances [4,6,7,21,22,23,24,25,36].

### 2.3. Statistical Analysis

All statistical analyses were performed in the SPSS software package (SPSS version 16.0, IBM, Armonk, NY, USA). Paired *t*-tests were performed to determine if there were any significant differences between fast and moderate speeds on the test variables (i.e., peak angles, angular velocities, GRF and joint contact forces). Statistical significance was set at *p* = 0.05. Cohen’s effect method (Cohen’s d) was used to calculate effect size. The value < 0.2 indicates a small effect size, close to 0.5 indicates a moderate effect size, and >0.8 indicates a large effect size.

## 3. Results

### 3.1. Joint Kinematics and GRF Variables

Figure 2 shows the hip, knee and ankle joint angles across time in both the fast and moderate movement conditions. The paired *t*-tests (Table 1, Figure 2) indicated that the fast movement condition had a significantly larger ankle dorsiflexion for all events (i.e., at initial contact, min, peak and take-off, *p* < 0.05; *d* > 0.8) but smaller peak pronation (*p* = 0.001; *d* > 0.8) and pronation at take-off (*p* = 0.020; *d* > 0.8) than the moderate movement condition. In addition, the fast movement condition demonstrated smaller peak and initial knee extension angles (*p* < 0.001; *d* > 0.8) and smaller hip peak extension (*p* = 0.007; *d* > 0.8) and extension at take-off (*p* < 0.001; *d* > 0.8) than the moderate movement condition (Table 1, Figure 2).

Figure 3 shows the hip, knee and ankle angular velocities during the fast and moderate movements. Compared to moderate movement (Table 2, Figure 3), the fast movement led to a higher knee extension but smaller hip extension velocities at initial contact (*p* < 0.001; *d* > 0.8), higher peak knee extension velocity (*p* < 0.001; *d* > 0.8), as well as higher knee flexion (*p* < 0.001; *d* > 0.8) and ankle plantarflexion at take-off (*p* = 0.033; *d* > 0.8).

For the GRF variables (Table 3), the fast movement condition had significantly higher first vertical (*p* = 0.005; *d* > 0.8) and horizontal (*p* = 0.007; *d* < 0.8) GRF peaks than the moderate movement condition. Additionally, the fast movement indicated a significantly higher second vertical GRF peak (*p* = 0.001; *d* > 0.8) but smaller second horizontal peak (*p* = 0.010; *d* > 0.8) than the moderate movement.

### 3.2. Joint Contact Force Variables

Figure 4 shows the medial–lateral and proximal–distal contact forces of the ankle, knee and hip joints. Compared to the moderate movement condition (Table 4), the fast movement produced a higher peak knee lateral contact force (*p* = 0.008; *d* > 0.8) and ankle lateral contact force (*p* < 0.001; *d* < 0.8). Additionally, the fast movement generated a higher first peak knee and hip distal contact forces (*p* < 0.001; *d* > 0.8), higher peak ankle distal contact force (*p* = 0.001; *d* > 0.8), higher valley knee and hip distal contact force (*p* = 0.001; *d* > 0.8) as well as higher second peak knee distal contact force (*p* = 0.001; *d* > 0.8) than the moderate movement condition (Figure 4 and Table 4).

## 4. Discussion

Through an approach using musculoskeletal modelling, this study examined the differences in ground reaction forces and lower limb joint contact forces during the take-off step in forehand overhead strokes with fast and moderate movements. The key results are summarized as follows: (1) Fast speed movement led to larger ankle dorsiflexion and knee extension velocity but smaller hip extension velocity at initial contact. (2) Fast movement displayed smaller peak ankle pronation, knee flexion and hip flexion but higher peak ankle dorsiflexion velocity and peak knee extension velocity. (3) Fast movement exhibited larger ankle dorsiflexion; smaller ankle pronation and hip extension angles; and higher plantarflexion, knee flexion and hip flexion velocities during take-off. (4) Fast movement induced larger first and second vertical peaks, larger first horizontal peak but lower second horizontal peak. (5) Fast movement led to higher peak ankle lateral and distal contact forces, higher knee lateral and distal contact forces, and higher hip distal contact forces than moderate movement conditions.

### 4.1. GRF, Joint Kinematics and Contact Forces in Braking Phase (Initial Contact, Peak)

Larger first vertical and horizontal GRF peaks were found in the fast movements. Exposure to repetitive and rapid overhead strokes places higher loads on the lower extremities of the athletes, resulting in higher lower extremity injuries [43]. Additionally, the hip joint angle variation changed little when the participants completed their overhead stroke with faster movement speeds. The fast speed movement condition had a larger peak hip distal force than the moderate, which suggested a fast force transmission along the direction of motion [44,45]. The hip joint appears to be an important technical factor influencing forehand overhead strokes and injuries during early landing period. At the knee joint, the fast movement condition showed a larger maximum knee angular velocity, which may relate to a larger knee distal (+2.8%) and lateral (+8.8%) contact forces during fast movement. The increase in joint contact force and joint velocity is attributed to the higher risk of knee joint injury [46], which shows a strong association between the joint contact force and the knee angular velocity. Therefore, reducing contact force and/or knee angular velocity would prove important in order to protect knee joints from injuries.

The foot in fast movements changed from pronation to supination, but the foot in moderate movements only showed decreasing pronation angle across the stance phase. The inconsistent change in coronal foot motion resulted in different forces acting on the ankle, for which fast movement condition had a higher ankle distal (+16.6%) and lateral (+18.4%) contact forces than the moderate movements. The higher ankle joint contact force is related to more ankle injuries [47]. However, it can be seen from Figure 4 that both the horizontal GRF and medial–lateral ankle contact force in the fast movement condition decreased faster than those in the moderate movement condition, indicating that the angle changing from pronation to supination could have delayed the action time of the lateral contact force. This is in line with a previous running study that argued that overpronation at the early stance phase of running delayed the peak of vertical GRF [48]. Considering the special physiological structure of ankle joint, strengthening exercises should be considered to protect the medial and lateral ligaments to prevent injury. While the longitudinal impact on injury risks for athletes is uncertain, the findings from the present study confirm that fast movements induce higher impact forces during forehand overhead stroke activities.

### 4.2. GRF, Joint Kinematics and Contact Forces in Propulsion Phase (Take-off, Min)

When performing a step back followed by vertical jump for an overhead stroke, participants may experience higher horizontal backward inertia prior to the take-off event. Athletes performing overhead strokes in the fast movement manner induced larger second vertical GRF peaks but smaller second horizontal GRFs than in the moderate movement condition, suggesting that higher efficiency is required to transfer horizontal inertia to the vertical force component. The range of hip motion was generally smaller in the fast movement condition compared with moderate movement condition. The fast movement condition appears to complete the hip movement more stably, resulting in a higher hip proximal contact force. In other words, the faster the movement speed, the higher the contact forces acting on the hip.

The range of knee angular velocity and the proximal–distal contact force were generally larger in the fast movement condition than in the moderate movement condition. In the running literature, kinematic variability of the lower limbs during movements has been linked to injury occurrences [49,50]. It is argued that lower variability may be indicative of localized mechanical stress on a certain body and higher variability may help re-distribute the load and, hence, prevent overuse injury [50]. This may imply that knee joint could be the key joint in the take-off. The knee medial–lateral force generally declined with contact time, which indicated that knee joint would play an important role in pushing the body in an upward movement [51]. It is also consistent with the human body structure to ensure the stability of the knee joint to produce effective force transmission upward. Liu et al.’s study [52] found that different take-off postures and techniques led to different force generation, and the knee bending angles were the key factor in generating an upward force. It was consistent with the conclusion that higher ranges of knee motion were required to generate higher/rapid jumping forces in the fast movement condition than moderate movements [53]. In a sprinting study, Liu (2008) showed that the GRF passed through the front of the joint center of knee before the instant of sprint take-off. The hind leg muscle produced significant bending torque in order to counteract this moment, which may increase the likelihood of an injury of the hind leg muscle. Such rapid/strenuous action was similar to the leg take-off movement in our study; therefore, more attention should be paid to hind leg mechanics to prevent sports injuries.

No observed differences were found in the ankle joints during take-off, but higher ankle angular velocity was found in the fast movement condition than the moderate movements. The foot changed from supination to pronation in the fast movement condition; it only increased the pronation amplitude in the moderate movement condition, and the ankle distal contact force was larger in the fast speed condition than that in the moderate movements. This indicates that the movement speed caused a larger downward ankle contact force, which was beneficial for athletes in order to perform rapid/effective jumps.

### 4.3. Experimental Limitation

Although this pilot study provided a theoretical basis for the lower limb mechanics during forehand overhead strokes, several limitations should be considered. First, a larger sample size with various genders and playing levels would be needed for a more comprehensive analysis to generalize the take-off joint mechanics amongst athletes. Second, the isolated and preplanned movement trials were measured in the laboratory. In realistic badminton game plays, the movements are rather unanticipated and consecutive to the preceding movements. The information collected from studying realistic movements would be valuable in sport training and performance. Third, we did not measure electromyography as well as upper limb and racket mechanics. Future studies could examine the relationship among racket, upper and lower limb mechanics to underlying mechanisms and strategies associated with the faster and slower overhead stroke movements.

## 5. Conclusions

Assessing different movement speeds may be valuable in analyzing sporting performances. With the increase in movement speed, larger joint contact forces were determined at all hip, knee and ankle joints. However, distinct joint angles and velocities were exhibited among the hip (smaller angle-larger velocities), knee (larger velocities) and ankle, suggesting that different joint coordination strategies should be utilized to achieve higher movement speeds. The larger joint contact forces possibly contributed to higher pressure on joint stability and muscle forces. Although fast movement speed of the forehand overhead stroke is always encouraged in offensive play, the information on take-off characteristics could be useful in predicting the performances and potential risks of hip, knee and ankle joints and could be insightful for designing training regimes.

## Figures and Tables

**Figure 1 biology-11-01401-f001:**
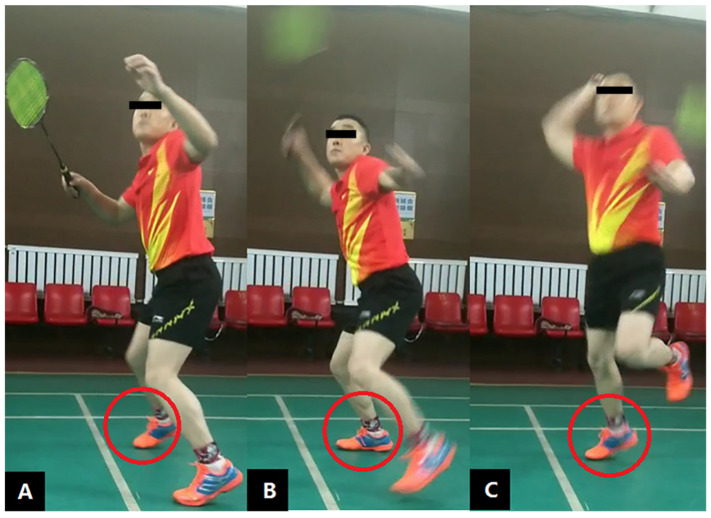
The sequence for the take-off step during a forehand overhead stroke: (**A**) right foot initially making contact with the ground, (**B**) peak GRF in the braking phase and (**C**) instant of take-off.

**Figure 2 biology-11-01401-f002:**
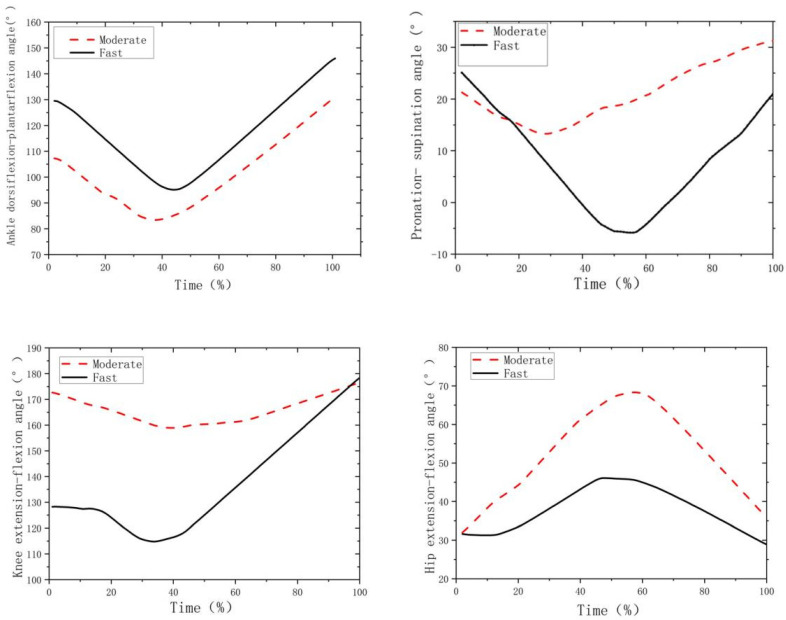
The curves of hip, knee and ankle joint angles across time.

**Figure 3 biology-11-01401-f003:**
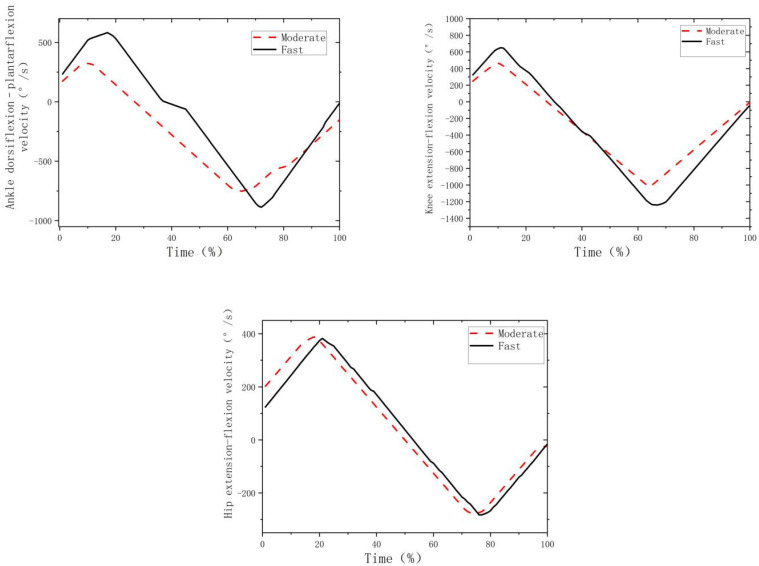
The curves of hip, knee and ankle angular velocities across time.

**Figure 4 biology-11-01401-f004:**
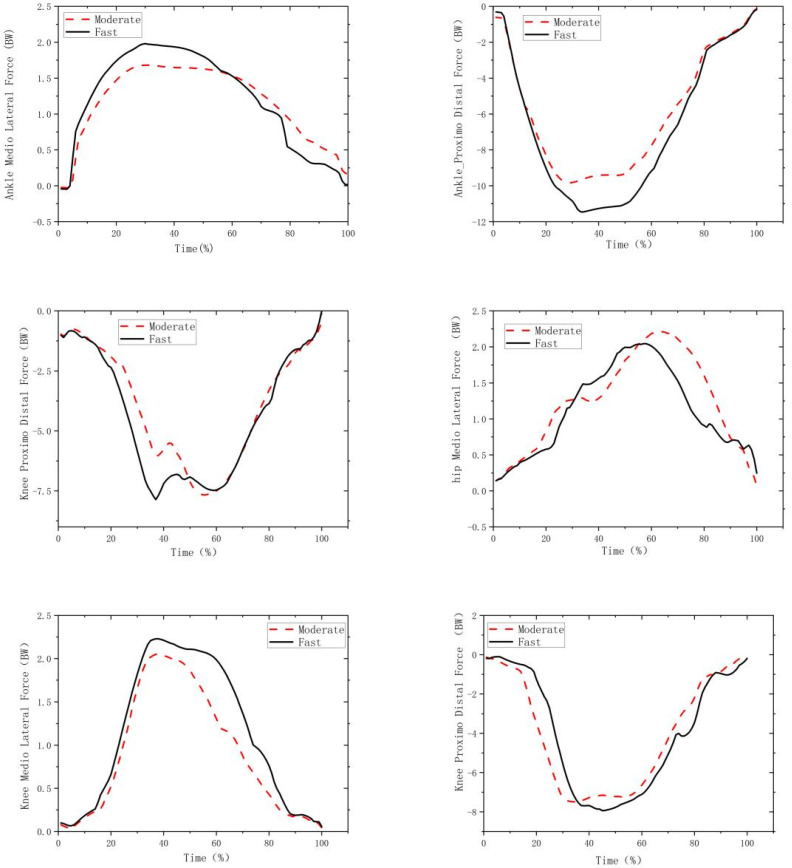
The curves of the medial–lateral (left) and proximal–distal (right) contact forces of ankle (upper), knee (middle) and hip (bottom) joints across time.

**Table 1 biology-11-01401-t001:** The hip, knee and ankle joint angles (°) in each event.

Variable	Event	Fast	Moderate	*Cohen’s d*
Ankle dorsiflexion–plantarflexion angle	Initial contact	128.26 ± 5.38 **	107.30 ± 2.83	4.87
	Min	94.30 ± 1.77 **	81.61 ± 3.77	4.30
	Take-off	145.95 ± 4.45 **	131.15 ± 6.26	2.72
Pronation–supination angle	Initial contact	25.11 ± 6.15	21.32 ± 5.30	0.66
	Min	−5.86 ± 3.39 **	13.29 ± 2.54	6.39
	Take-off	21.62 ± 4.21 *	31.39 ± 4.62	2.21
Knee extension–flexion angle	Initial contactMinTake-off	128.26 ± 5.38 **	172.70 ± 4.65	8.83
		114.78 ± 11.21 **	159.13 ± 9.14	4.33
		178.37 ± 1.24	176.6 ± 1.05	1.54
Hip extension–flexion angle	Initial contactPeakTake-off	31.61 ± 5.05	32.00 ± 3.52	0.08
		46.09 ± 6.38 **	68.32 ± 3.21	4.40
		28.5 ± 6.76 **	34.89 ± 3.22	1.20

Dorsiflexion (+)/plantarflexion (−); pronation (+)/supination (−); extension (+)/flexion (−). * represents significant difference with *p* ≤ 0.05; ** represents significant difference with *p* ≤ 0.01.

**Table 2 biology-11-01401-t002:** The hip, knee and ankle angular velocities (°/s) in each event.

Variable	Event	Fast	Moderate	*Cohen’s d*
		Mean	Mean	
Ankle dorsiflexion–plantarflexion velocity	Initial contact	233.80 ± 40.74	173.59 ± 52.57	1.28
	Peak	582.02 ± 177.225 **	323.24 ± 6.19	2.06
	Min	−886.967 ± 104.937 *	−752.22 ± 45.375	1.66
Knee extension–flexion velocity	Initial contact	324.09 ± 44.23 *	248.16 ± 23.38	2.14
	Peak	651.08 ± 89.71 **	465.78 ± 19.85	2.85
	Min	−1240.31 ± 75.98 **	−1016.00 ± 49.58	3.49
Hip extension–flexion velocity	Initial contact	123.93 ± 24.28 **	202.36 ± 27.06	3.05
	Peak	382.44 ± 35.69	387.79 ± 48.99	0.12
	Min	−282.81 ± 46.49	−277.34 ± 57.57	0.10

Dorsiflexion (+)/plantarflexion (−); extension (+)/flexion (−). * represents significant difference with *p* ≤ 0.05; ** represents significant difference *p* ≤ 0.01.

**Table 3 biology-11-01401-t003:** Vertical and horizontal GRF data (BW).

Variable	Event	Fast	Moderate	*Cohen’s d*
Vertical GRF	First peak	2.16 ± 0.14 **	1.59 ± 0.17	3.66
	Valley	1.49 ± 0.06	1.32 ± 0.11	1.91
	Second peak	2.29 ± 0.03 **	1.87 ± 0.05	10.18
Horizontal GRF	First peak	1.54 ± 0.21 **	1.43 ± 0.29	0.43
	Valley	1.26 ± 0.20	1.14 ± 0.17	0.64
	Second peak	1.18 ± 0.24 **	1.45 ± 0.14	1.37

** represents significant difference with *p* ≤ 0.01.

**Table 4 biology-11-01401-t004:** The hip, knee and ankle medial–lateral and proximal–distal contact forces (N/BW) in each analyzed event.

Variable	Event	Fast	Moderate	*Cohen’s d*
Ankle medial–lateral force	Peak	1.99 ± 0.34 **	1.68 ± 3.95	0.11
Ankle proximal–distal force	Peak	−11.46 ± 0.48 **	−9.83 ± 0.13	4.63
Knee medial–lateral force	Peak	2.23 ± 0.05 **	2.05 ± 0.05	3.6
Knee proximal–distal force	First peak	−7.70 ± 0.64 **	−7.49 ± 0.26	0.42
	Valley	−7.68 ± 0.26 **	−7.16 ± 0.18	2.32
	Second peak	−7.94 ± 0.79 **	−7.23 ± 0.47	1.09
Hip medial–lateral force	Peak	2.04 ± 0.20	2.21 ± 0.13	1.00
Hip proximal–distal force	First peak	−7.86 ± 0.15 **	−6.02 ± 0.18	11.10
	Valley	−6.81 ± 0.86 **	−5.50 ± 0.56	1.80
	Second peak	−7.48 ± 0.24	−7.66 ± 0.21	0.79

Medial (−)/lateral (+); proximal (+)/distal (−). ** represents significant difference with *p* ≤ 0.01.

## Data Availability

The data presented in this study are available from the corresponding author upon reasonable request.

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
