# Peer review of "Biomechanical Analysis on Skilled Badminton Players during Take-Off Phase in Forehand Overhead Strokes: A Pilot Study"

_biology, 2022, doi:10.3390/biology11101401_

Round 1
Reviewer 1 Report
Dear authors,
First of all,
I would like to congratulate the authors for the chosen theme, in a sport that is not as studied as others. A few comments have been included for the improvement of the manuscript.
Tittle: The title is fine, but since it is a study with such a small sample, it should be included that it is a pilot study.
Line 74: include the manufacturer information
The sample is very small and should be enlarged, why didn't more participants take part?
Line 100-103: Please, specify how this anatomical point were determined
Procedure, line 127-143. The procedure is explained in a very similar way to the article Joint contact force and movement deceleration among badminton forward lunges: a musculoskeletal modelling study. Perhaps the authors should paraphrase this paragraph a little more to avoid plagiarism.
Results: The p-value is usually represented by lowercase p in italics.
I don't know what format the figures are in, but the quality could be improved.
Author Response
Reviewer 1
First of all, I would like to congratulate the authors for the chosen theme, in a sport that is not as studied as others. A few comments have been included for the improvement of the manuscript.
- Tittle: The title is fine, but since it is a study with such a small sample, it should be included that it is a pilot study.
Reply: Thank you for the suggestion. We are now added “A pilot study” in the title as suggested.
- Line 74: include the manufacturer information.
Reply: Thank you for the suggestion. The manufacturer information has been removed as suggested (line 77).
- The sample is very small and should be enlarged, why didn't more participants take part?
Reply: Thank you for the comment. Our manuscript reported the overhead stroke characteristics of the very top elite athletes. Our participants were all national first-level athletes and have trained regularly for at least ten years, which is limited number and resulted in small sample sizes. In addition, the sample size included in this study was calculated using Gpower software, which supported our sample size is matched with the minimum sample requirements(line 88).
- Line 100-103: Please, specify how this anatomical point were determined.
Reply: The anatomical points were identified by the same experienced experimenter. Then, the points are to process the marker position in the Cartesian coordinate system as a rotation angle according to the anatomical posture of the human body. The joint angular displacements of the hip and knee joints in the sagittal, frontal and transverse planes were calculated. (Line 99-108)
- Procedure, line 127-143. The procedure is explained in a very similar way to the article Joint contact force and movement deceleration among badminton forward lunges: a musculoskeletal modelling study. Perhaps the authors should paraphrase this paragraph a little more to avoid plagiarism.
Reply: Thank you for the suggestion. We have now re-phrased the paragraph as suggested (Line 137-154).
Results: The p-value is usually represented by lowercase p in italics.
Reply: Thank you for the suggestion. All p-value is now revised as “p” as suggested.
I don't know what format the figures are in, but the quality could be improved.
Reply: The figures have been improved.
Reviewer 2 Report
The article focuses on an exciting topic. There are, however, some amendments to be made before acceptance.
The English should be deeply revised.
The level of the players should be described. Having 10-13 yrs of practice is not enough to understand their competitive level.
L118 sd is missing.
What does this mean? “The movement speed conditions were counterbalanced across participants.”.
No reference was made for the synchronization between 2 such different acquisition frequencies (200 vs. 1000-Hz).
“A total of six trials were measured in each of the speed conditions.” Per participant, or overall? If the latter, does it means that only one trial was considered? Which one? How was individual variability considered?
With such a sample size, the p-value is insufficient to make such conclusions. Effect size is mandatory.
Finally, and most importantly, it is not clear what the study’s relevance is. Authors state, “Assessing different movement speeds may be valuable in sporting performances.” Please elaborate on that. Does it mean that in the Olympic game, we should tell the athlete not to perform so fast or he can be injured?
Author Response
Reviewer 2
The article focuses on an exciting topic. There are, however, some amendments to be made before acceptance.
- The English should be deeply revised.
Reply:
- The level of the players should be described. Having 10-13 yrs of practice is not enough to understand their competitive level.
Reply: Thank you for the suggestion. We are now provided the detailed description for the competitive level. The corresponding statements are now read: “They were competitive club-level badminton players with regular training at least ten years (ranged from 10 to 13 yrs). The participants are all national first-level badminton players for China” (Line 90-92).
- L118 sd is missing.
Reply: Thank you for the comments. The missing SD is now provided. (Line 127-128)
- What does this mean? “The movement speed conditions were counterbalanced across participants.”.
Reply: Thank you for clarifying this. In order to avoid the learning effect, we assigned half of the participants with moderate speed condition followed by fast speed condition and another half of the participants with fast speed condition followed by moderate speed condition. The corresponding statement is now read: “The order of movement speed conditions presented were counterbalanced across participants” (line 128-129)
- No reference was made for the synchronization between 2 such different acquisition frequencies (200 vs. 1000-Hz).
Reply: Thank you for asking this. To our best knowledge, almost all biomechanical studies have different sampling frequencies between different modalities (e.g., force plates, pressure plate, motion capturing system, EMG system). In our study, we firstly synchronized both motion capturing system and force plate (i.e., both system started at the same time) for all data collection. Then, it is commonly performed the down-sample process from the higher frequency modality (2000 Hz, force plate) into lower frequency (200 Hz, averaged every 5 data points into 1 data point).
- “A total of six trials were measured in each of the speed conditions.” Per participant, or overall? If the latter, does it means that only one trial was considered? Which one? How was individual variability considered?
Reply: In this study, each participant performed three successful trials for each of the two motion speeds, resulting in a total of six trial for each participant. (line 129-131).
- With such a sample size, the p-value is insufficient to make such conclusions. Effect size is mandatory.
Reply: Thank you for the suggestion. We added Cohen’s d values for the effect size verification. The corresponding statement is now read:“Cohen's effect method (Cohen'sd ) to calculate effect size. The value <0.2 indicates a small effect size, close to 0.5 indicates a moderate effect size, and >0.8 indicates a large effect size.” (Line 163-165)
- Finally, and most importantly, it is not clear what the study’s relevance is. Authors state, “Assessing different movement speeds may be valuable in sporting performances.” Please elaborate on that. Does it mean that in the Olympic game, we should tell the athlete not to perform so fast or he can be injured?
Reply: 1) The purpose of this study is to evaluate the load of the lower extremity joints at different speeds. According to the results in the text, the performance and injury risk of each joint can be further observed.
2) Evaluating the value of different exercise speeds for exercise performance is to find the optimal exercise mode.
3) Predict the potential risk of each joint through the information of the take-off characteristics and achieve higher sports performance with the coordinated movement of the joints, instead of giving recommendations through a single indicator.
Round 2
Reviewer 1 Report
The authors have made the suggested changes so I accept the manuscript in the current format.
Reviewer 2 Report
Good job improving the quality of the manuscript.